# Microwave-Assisted Green Synthesis and Antioxidant Activity of Selenium Nanoparticles Using *Theobroma cacao* L. Bean Shell Extract

**DOI:** 10.3390/molecules24224048

**Published:** 2019-11-08

**Authors:** Cristina Mellinas, Alfonso Jiménez, María del Carmen Garrigós

**Affiliations:** Department of Analytical Chemistry, Nutrition & Food Sciences, University of Alicante, 03690 San Vicente del Raspeig, Alicante, Spain; cristina.mellinas@ua.es (C.M.); alfjimenez@ua.es (A.J.)

**Keywords:** *Theobroma cacao* L. bean shell extract, selenium nanoparticles, microwave-assisted synthesis, central composite design, antioxidant

## Abstract

Selenium nanoparticles (SeNPs) are successfully synthesized through microwave heating by using *Theobroma cacao L.* bean shell extract as a stabilizing and capping agent. Response surface methodology is used to obtain optimal synthesis conditions. The effect of microwave power, irradiation time and amount of Na_2_SeO_3_ are evaluated on crystalline size by X-ray Diffraction (XRD) and *Z*-potential by Dynamic Light Scattering (DLS) using a central composite design (CCD). Optimal synthesis conditions are determined as 15.6 min, 788.6 W and 0.14 g of sodium selenite using 50 mL of *Theobroma cacao* L. bean shell extract. The successful biosynthesis of SeNPs is confirmed by UV-visible and Fourier Transformed Infrared (FTIR) spectroscopic analyses. The XRD pattern and Raman spectra show the presence of trigonal and amorphous synthesized SeNPs. Spherical SeNPs are observed by Transmission Electron Microscopy (TEM) with a particle size of 1–3 nm in diameter, at least one order of magnitude lower than those previously reported. The obtained SeNPs can be stable up to 55 days at 4 °C. Additionally, the SeNPs show an excellent antioxidant performance by the 2,2′-azino-bis(3-ethylbenzothiazoline-6-sulphonic acid) (ABTS) and ferric reducing antioxidant power (FRAP) methods, with potential application in different sectors, such as food, medical and pharmaceutical.

## 1. Introduction

Selenium is an essential element for the operation of multiple biological processes in humans, but their concentration threshold between functionality and toxicity is very narrow [1]. Selenium can be present in Nature showing different polymorphic structures, either crystalline or amorphous. The crystalline forms include three allotropes of monoclinic selenium (m-Se) containing rings of Se_8_ with different packing to give red monoclinic forms (α, β and γ). Black trigonal selenium (t-Se) is the most stable crystalline form at room temperature [2]. The most important non-crystalline forms of selenium are the red amorphous (a-Se), black amorphous and vitreous selenium [3].

Since the late 1990s, the application of nanotechnology in different fields has received extensive attention. The formulation of “tailor-made” nanomaterials could result in highly biocompatible and biodegradable composites with potential applications in the medical, photocatalytic, food packaging and electronic sectors [4,5,6,7]. Specifically, selenium nanoparticles (SeNPs) have been reported to exhibit excellent bioavailability, low toxicity and strong antioxidant and antibacterial activity [8].

The development of green and sustainable synthesis methods of nanoparticles has been the object of research in the last years [9] and, particularly, plant extracts have been reported as potential precursors for the synthesis of metallic nanoparticles [10]. Many methods for the synthesis of SeNPs, by using chemical, biological and physical approaches, have been proposed [11,12]. Nevertheless, some of them involve complicated, tedious and non-sustainable protocols. Therefore, the introduction of synthesis methods based on the use of plant-derived extracts is a green, single-step, eco-friendly, bio-reductive and cost-effective approach, requiring lower reaction times and solvent use. Several parameters, such as the metal salt concentration, the use of suitable reducing agents, temperature, pH and reaction time, are critical to obtain high yields and purities in the synthesis of SeNPs [13]. Different strategies in the synthesis of SeNPs involve the use of ascorbic acid [14] or polyphenols derived from plants [15] as reducing agents. However, SeNPs obtained in that way show some limitations, such as their poor stability, so the addition of stabilizing agents during synthesis becomes necessary. Recent studies have reported that the functionalization of SeNPs with natural polysaccharides may improve effectively final nanoparticle stability, giving to these polysaccharide-conjugated SeNPs the desired properties [1,16,17]. Different plant extracts containing bioactive compounds, which can be used in synthesizing and stabilizing SeNPs, have been studied as response surface methodology (RSM) successfully applied to develop empirical models for the prediction of the SeNPs synthesis conditions [18,19]. RSM is the most adequate statistical tool to obtain optimal synthesis conditions by reducing the number of experiments and evaluating the interaction effects of the independent factors on the response variables [20].

Microwave-assisted synthesis has been studied extensively recently due to its advantages over traditional heating methods [21] for the synthesis of metallic nanoparticles. Microwave radiation can be absorbed easily by the mixture solution containing the ions salt and reducing agents. Consequently, localized superheating occurs resulting in fast and efficient heating, completing reactions in minutes or even seconds using less energy [22,23]. Green synthesized NPs under microwave irradiation have been reported to have minimum particle size and particle size distributions, and maximum stability [19]. These processes are influenced by many factors, such as time, temperature, microwave power, stirring rate, precursor-reducing agent ratio, pH or solvent [24].

The aim of this work is the optimization of a green microwave-assisted synthesis method for SeNPs using *Theobroma cacao* L. bean shell (CBS) extract as bio-reductant and capping agent. Cocoa (*Theobroma cacao* L.) is the name given to the fruit of the cocoa tree and its seeds are commonly called cocoa beans. Cocoa shell is the major industrial by-product of the cocoa bean obtained during processing into chocolate and other products. CBS has been reported as a rich source of bioactive compounds, including polysaccharides, proteins and phenolics [25], with potential applications for the synthesis of nanoparticles, but not used before for this purpose to the best of our knowledge. Experimental synthesis parameters are optimized through RSM using a central composite design (CCD) to obtain small-sized and stable SeNPs, which were fully characterized by different analytical techniques. The stability and antioxidant properties of the synthesized SeNPs with time also were studied to assess their potential as antioxidant materials in the food industry and other fields.

## 2. Results and Discussion

### 2.1. Quantitative Analysis of Cocoa Bean Shel Extract (CBSE)

The quantitative analysis of CBSE showed high contents of uronic acid (related to pectin content), polyphenols, polysaccharides and proteins (Table 1). These active components may be combined with the metal precursor to provide the nanoparticles. Specifically, polyphenols (possessing hydroxyl reducing groups) and proteins may play a key role in reducing selenium ions to their element and stabilizing the formed SeNPs, respectively [18,19]. Hydroxyl groups also may act as capping agents by the deposition of bioactive compounds on the SeNPs’ surface [8] preventing them from aggregation and, also, adding antioxidant capacity. The concentration of polysaccharides present in natural extracts also has been reported to influence the size of the obtained nanoparticles [26,27]. The use of CBSE allows the reduction of toxic chemicals, leading to a green synthesis of SeNPs.

### 2.2. Optimization of Synthesis Conditions of SeNPs

A central composite design (CCD) strategy was used to optimize SeNPs synthesis conditions with 23 runs which were performed randomly. The design matrix and results obtained for all experiments are shown in Table 2. The influence of three independent variables, including microwave power, irradiation time and amount of Na_2_SeO_3_ on two responses (crystalline size and *Z*-potential), was evaluated. All the studied responses were expressed as a function of significant independent variables (*p* < 0.05) by using second-order polynomial equations as follows:Size = 36.84 − 60.43 × C + 202.29 × C^2^(1)
Zeta Potential = 64.12 − 1.59 × A − 93.11 × C − 3.82 × A × C + 0.00 × B^2^ + 261.66 × C^2^(2)
where A, B and C represent extraction time, microwave power (W) and sodium selenite amount, respectively.

Analysis of variance (ANOVA) was carried out to analyse the effect of the studied variables on the selected responses and to evaluate the reliability of the fitted models (Table 3). The coefficient of determination (R^2^) reflects the proportion of variation in the response attributed to the model rather than to random error. High values were obtained for R^2^ of the models (0.9121 and 0.9327 for crystalline size and *Z*-potential, respectively), with adjusted R^2^ values quite close to R^2^, confirming the accuracy of the fitted models in correlating the results with the experimental data. The high *p*-values shown for lack of fit (0.1215 and 0.1558 for crystalline size and *Z*-potential, respectively) also indicated that the lack of fit was not significant (*p* > 0.05), confirming the good fitness of the generated models. Finally, the low values of coefficients of variation (CV) < 3% suggested high reproducibility of results and the reliability of the models.

The optimal SeNPs synthesis conditions were calculated to minimize nanoparticle crystalline size and maximize *Z*-Potential values using a simultaneous optimization by means of a desirability function (D) which searches a combination of the different variables levels that satisfies all the requirements for each response at the same time [28]. The value obtained for D was one, giving results that fully satisfied both variables. Accordingly, the optimal SeNPs synthesis conditions were: 15.6 min, 788.6 W and 0.14 g of sodium selenite using 50 mL of CBSE solution.

Three confirmation experiments were conducted under optimal synthesis conditions to confirm the suitability of the model equations. The SeNPs obtained showed experimental values of 41.5 ± 1.7 nm of diameter and −28.6 ± 5.3 mV for *Z*-Potential, which did not differ significantly (*p* > 0.05) from predicted values of 40.8 ± 5.4 nm and −27.1 ± 4.4 mV, respectively. These results showed the strong correlation between the predicted and experimental data, confirming that the models were reliable for the optimization of green SeNPs synthesis through microwave heating.

#### 2.2.1. Effect of Synthesis Variables on SeNPs Crystalline Size

Biomolecules present in plant extracts such as polysaccharides, proteins, flavonoids, terpenoids, alkaloids, polyphenols and vitamins are involved in bio-molecular reduction, formation and stabilization of SeNPs and they could have a better control over the size and the shape of the nanoparticles being synthesized [29]. Optimal synthesis conditions were focused on obtaining small-sized SeNPs, which have been reported to be more effective in different applications [11,12,30]. The size of crystalline synthesized SeNPs was evaluated by means of XRD. ANOVA results (Table 3) showed that the amount of sodium selenite had a great effect on the synthesis of SeNPs with a linear and square effect (*p* < 0.001). A smaller size of SeNPs was obtained by using low concentrations of the metal precursor. Moreover, it was observed that higher amounts of sodium selenite increased the crystalline size of SeNPs, at a fixed concentration of CBSE. This behaviour could be explained by considering that high amounts of the metal precursor involve a large number of selenium nuclei that could agglomerate during the growth step, giving rise to nanoparticles with larger size [23]. These results agree with those reported by other authors who evaluated the influence of using reducing and capping agents during the synthesis of SeNPs [31,32], obtaining larger nanoparticles when the concentration of these agents was lower than the concentration of the metal precursor.

Conversely, time and microwave power did not show a significant effect (*p* > 0.05) on SeNPs crystalline size, under the studied conditions, probably due to the great effect of the metal precursor. To conclude, to obtain small-sized SeNPs, low amounts of the metal precursor are needed, regardless of the time and conditions of the microwave heating used.

#### 2.2.2. Effect of Synthesis Variables on *Z*-Potential Values

The stability of nanoparticles is an important feature for their application in different fields. The *Z*-potential is an electrochemical parameter related to the mobility of ions present in a solution. Regarding *Z*-potential measurements, an electrical field is applied across the solution containing the nanoparticles and the movement caused by the electrophoretic mobility is measured. The *Z*-potential is indicative of the stability of a particle since it measures the potential required to penetrate the layer of ions close to a particular particle to destabilize it [33,34]. Therefore, and considering the importance of getting stable SeNPs, synthesis conditions were optimized to obtain nanoparticles with high *Z*-potential values.

Seen in Table 3, all synthesis variables (i.e., amount of metal precursor, time and microwave power) had a significant effect (*p* < 0.05) on the stability (*Z*-potential) of SeNPs. The obtained SeNPs were well stabilized by biomolecules present in CBSE which can be deposited onto the nanoparticle’s surface, preventing aggregation of the grown nanoparticles and resulting in more stable SeNPs [31,35]. The amount of metal precursor showed a linear (*p* < 0.01) and square (*p* < 0.001) significant effect on *Z*-potential. The ratio between CBSE and sodium selenite showed the same trend as in the case of the nanoparticles crystalline size. So, when the concentration of sodium selenite is low, the active compounds present in CBSE could help to decrease the attractive forces between the SeNPs produced, resulting in greater stability in solution [26,27].

A significant interaction (*p* < 0.001) between reaction time and sodium selenite amount on *Z*-potential was observed (Figure 1). The *Z*-potential value decreased with increasing reaction time and sodium selenite amount. This probably is due to the increase of the free energy of the system that favours the aggregation of nanoparticles and, consequently, their stability is reduced. However, when the time was increased and the precursor amount decreased, SeNPs showed higher stability, which was attributed to a higher separation between the selenium nuclei thanks to the stabilizing agents present in CBSE, reducing the probability of aggregation during the formation of SeNPs [36,37,38].

Regarding microwave power, it showed a great effect on the *Z*-potential of SeNPs with a high significant square effect (*p* < 0.001) (Table 3). Thus, high microwave power resulted in highly stable SeNPs. The use of microwave irradiation increased the reaction kinetics with rapid initial heating, resulting in enhanced reaction rates [21]. Consequently, the high formation rate of SeNPs decreases the probabilities of aggregation since they can be stabilized quickly by the CBSE compounds.

The obtained results indicate that CBSE can be used effectively in the synthesis of SeNPs to obtain small crystalline particles with high stability, which could be related to the strong reductant and stabilizing agents present in CBSE (Table 2).

### 2.3. Characterization of SeNPs Obtained under Optimal Synthesis Conditions

#### 2.3.1. XRD Analysis

The XRD pattern of synthesized SeNPs is shown in Figure 2A, where two different regions were observed. A broad shoulder between 10–30°, at low angles, was shown corresponding to the amorphous phase (a-SeNPs). This behaviour also was observed by other authors who reported the predominance of the amorphous region in synthesized SeNPs [39,40,41]. However, some peaks at higher diffraction angles 2θ (degrees) of 28.4, 31.6, 40.5, 45.5, 50.3, 56.6 and 66.6°, corresponding to different crystalline planes of SeNPs, were observed. These results are in agreement with those reported by Vieira et al. who ascribed peaks at similar angles to the 100, 101, 102, 111, 201, 112 and 103 crystalline planes [17]. This pattern could be indexed according to the trigonal phase of SeNPs (crystalline t-SeNPs), in agreement with results obtained by Raman spectroscopy. The presence of this structure could be due to the partial conversion of amorphous nanoparticles to their most stable crystalline form [23]. The prepared SeNPs calculated crystalline size was 41.5 nm using Debye–Scherrer’s equation, which was lower than the reported value of 58 nm by Alagesan et al. [8] using a leaf extract of *Withania somnifera*.

#### 2.3.2. Raman Spectroscopy

Raman spectroscopy is known to be sensitive to differences in various allotropic modifications and crystallinity of selenium in SeNPs [42] and empirical characteristic frequencies can be used for interpretation of the Raman spectra [43,44]. Figure 2B shows the Raman spectrum of SeNPs obtained under optimal conditions in the lower-frequency region. Two main peaks were observed at 236 and 251 cm^−1^, characteristic of trigonal [23] and amorphous [45] phases in SeNPs, respectively. Specifically, the band at 251 cm^−1^ is associated with the A1 stretching Se-Se mode [42], and the peak centered at 236 cm^−1^ is a characteristic signature of the symmetric stretching mode of t-Se, which can be attributed to the vibration of Se helical chains [46]. Similar results were observed by Zhang et al., who reported trigonal Se (t-Se) and monoclinic Se (m-Se) located at 234 and 254 cm^−1^, respectively [47]. The obtained results indicate a temporal intermediate of m-Se and t-Se in experimental synthesis conditions, supporting the formation of t-Se nanostructures, as already determined by XRD. Although the amorphous state is unstable and tends to transform into the more stable trigonal form, especially at high temperatures [48], this transformation was not completed under synthesis conditions maintaining m-Se in solution due to the stabilization of active compounds present in CBSE, which contributed to preserving the amorphous region in these SeNPs.

#### 2.3.3. TEM

The morphological analysis of SeNPs is very important to assess their homogeneity and further possibilities for industrial applications. TEM images of SeNPs obtained under optimal conditions are shown in Figure 3A at different magnifications. Synthesized SeNPs, as it can be observed, were mostly spherical and well dispersed, which demonstrated that the thermodynamic stable synthesized SeNPs had minimum surface energy, in agreement with the high value of the *Z*-potential obtained [18]. A uniform distribution was found for these nanospheres (Figure 3B) ranging in size from 1 to 3 nm in diameter, suggesting that crystalline forms were predominant in the SeNPs structure and indicating that SeNPs were well-stabilized by CBSE. The biological properties of SeNPs depend on their size having a greater activity with decreasing particle size [49]. Results obtained in this work clearly are advantageous over those reported by other authors, with diameters of amorphous SeNPs in the range of 50–500 nm [40,50,51,52,53], one or two orders of magnitude higher than results obtained in this study using CBSE. Liu et al. obtained SeNPs in the presence of polysaccharides extracted from *Catathelasma*
*ventricosum* with an average size of 50 nm [54], while similar values were obtained using carboxylic curdlans with various molecular properties [31]. Sharma et al. and Karuppannan et al. reported SeNP sizes ranging from 3 to 18 nm and 4 to 16 nm by using dried *Vitis vinifera* (Raisin) extract and *Diospyros montana* leaf extract, respectively [32,55]. So, it is clear that sizes and shapes of synthesized SeNPs are dependent on the natural extract used for the green synthesis method, which may be attributed to the differential reduction potential and capping ability of the phytoconstituents present in the extract used [32].

It particularly is relevant to compare our results with those obtained for biologically-synthesized SeNPs since some authors used bacteria to get homogeneous nanoparticles, but all of them obtained diameters higher than 10 nm by some agglomeration through the synthesis process [56,57,58]. To conclude, to the best of our knowledge, no reports have been found with sizes of synthesized SeNPs lower than 10 nm, as found in this work, showing the great potential of CBSE-mediated microwave synthesis to obtain SeNPs with low diameters and no agglomerated structures that could be related to strong reductant and stabilizing agents present in CBSE.

#### 2.3.4. DLS

The *Z*-potential obtained for SeNPs under optimal synthesis conditions was −28.6 ± 5.3 mV, indicating that the formed NPs were surrounded with negatively charged groups and had high stability [18]. Negative *Z*-potential values are related to the presence of negatively charged functional groups from the biomolecules present in CBSE at the NPs surface. This is the case of carboxyl and hydroxyl groups present in CBSE polysaccharides and polyphenols (Table 2), respectively, that can be used to bind SeNPs. When all particles in suspension have an overall negative *Z*-potential, as in this case, they would tend to repel each other, favouring to exist in dispersion form and resulting in a low trend to agglomeration, as already indicated by other authors [36]. So, the negative charged potential value was imparted to SeNPs due to the reducing agents (phenolics, polysaccharides, proteins) of CBSE.

Therefore, the sign of *Z*-potential gives information about the compounds that stabilize SeNPs, but the absolute value refers to the degree of stabilization under the studied conditions due to Van der Walls interactions [34]. It is well known that *Z*-potential absolute values higher than 30 mV result in highly stable nanoparticles [59,60] in this context. Results obtained in this study were comparable or even higher in absolute value (meaning higher stability) than those obtained by other authors who synthesized SeNPs under different approaches. Meng et al. obtained a *Z*-potential value of −20.39 mV [61], while Menon et al. reported −36 mV [36], Luesakul et al. indicated 14.1 mV [62] and finally −23.2 mV was the average *Z*-potential value obtained by Cui et al. [50], for instance. To conclude, the use of microwave-assisted synthesis supported by CBSE resulted in highly stable SeNPs.

#### 2.3.5. Optical Properties

UV-visible spectrophotometry was used to examine the formation of SeNPs using CBSE. Figure 4A shows a clear characteristic peak was obtained at 276 nm, indicating that CBSE has reduced and stabilized SeNPs formation. This peak is representative of Surface Plasmon Resonance (SPR) caused by the presence of conduction electrons in the SeNPs’ surface [63]. This result is in good agreement with those obtained by other authors, who reported the presence of a UV-Visible absorption maximum between 200–400 nm during the biosynthesis of SeNPs using extracts derived from plants [15,16,36,51,52].

The direct optical band gap for SeNPs obtained under optimal conditions was calculated from the optical absorption spectra by using the following equation:(αhυ)^n^ = A(hυ − E_g_)(3)
where E_g_ is the energy gap, α is the optical absorption coefficient, h is Planck’s constant, υ is the frequency of light, A is the constant of proportionality and *n* = 2 for direct band gap energy.

The band gap was obtained by extrapolating the straight portion of the graph (αhυ)^2^ versus (hυ) on the energy axis at α = 0. Regarding this plot (Figure 4B), the band gap energy was found to be 4.21 eV (R^2^ = 0.9989) for SeNPs obtained under optimal conditions using microwave assisted synthesis. This band gap energy for green synthesized SeNPs is higher than reported band gaps of 1.65, 1.85 and 2.05 eV for black, monoclinic and amorphous Se, respectively [64], and experimental values reported by other authors which ranged from 2.0 to 3.1 eV [8,23,65]. These differences in band gap energy could be related to the small nanoparticle size obtained in this work, as observed by TEM. Furthermore, when particle size is smaller than the Bohr excitation radius, the band gap will be enlarged due to the quantum confinement effect obtaining a blue shift of the band gap energy for SeNPs compared to its bulk counterpart [66]. So, the bigger the particle size, the smaller the band gap energy reported by other authors, as nanoparticles agglomeration is an important feature in previously reported works and not observed in this study. Therefore, it can be concluded that nanoparticle size has a major influence in the optical absorption spectra of SeNPs, resulting in higher band gaps for small-sized nanoparticles, as observed in our measurements [23,67].

#### 2.3.6. FTIR

FTIR analysis was used to obtain information about chemical compounds involved in the reduction and stabilization of SeNPs. Figure 5 shows the FTIR spectra of CBSE and SeNPs under optimal conditions. Characteristic absorption peaks were observed for CBSE, around 3280 cm^−1^ (OH stretching), 2934 cm^−1^ (CH stretching), 2854 cm^−1^ (carboxylic acid O–H), 1713 cm^−1^ (C=O stretching), 1648 cm^−1^ (amide I vibrations), 1210 cm^−1^ (C-O-C vibration in ester groups) and 1012 cm^−1^ (superposition in the plane of C-H bending of polysaccharides) [68]. These results indicate the presence of various functional groups as biomolecules in CBSE (Table 2), such as hydroxyl groups in polyphenols and amide groups in proteins having a key role in the reduction of selenium ions to their element and in the stabilization of the formed SeNPs, respectively [19].

However, the broad intense peak at 3280 cm^−1^ of CBSE was shifted to 3265 cm^−1^ for SeNPs, suggesting a strong hydrogen bonding interaction between selenium and the O-H groups from CBSE facilitating the biosynthesis of SeNPs through the formation of Se-O-H bonds. Similar phenomena were reported by other authors in SeNPs synthesis [27,35,40,69]. Additionally, the peak at 1713 cm^−1^ (carbonyl C=O stretching) of CBSE disappeared in SeNPs, which specify that carbonyl C=O stretching has enabled the synthesis of SeNPs in accordance to results reported by Gunti et al. for phytofabrication of SeNPs from *Emblica officinalis* fruit extract [70]. Similarly, the peak at 1648 cm^−1^ (amide I vibrations) in CBSE was shifted to 1615 cm^−1^ in SeNPs biosynthesis, showing the interaction of CBSE proteins with selenium through the amine groups. The typical polysaccharide vibration region (1210–1012 cm^−1^) also was shifted in SeNPs to higher frequencies. As a result, FTIR spectra of SeNPs showed the presence of various capping biomolecules from CBSE at the SeNPs surface, such as proteins and polysaccharides, that successfully have facilitated the biosynthesis of SeNPs by reduction processes and could aid in protection of SeNPs from aggregation, retaining their long-term stability, in line with reported literature on biosynthesized Se nanostructures [42,70]. Finally, the new peak observed at 807 cm^−1^ in SeNPs, and not present in CBSE spectra, was associated with Se-O stretching vibration, confirming the successful biosynthesis of SeNPs during the microwave-assisted synthesis process with CBSE, as already reported by Chen et al. [53].

#### 2.3.7. Stability of SeNPs and Antioxidant Activity

The stability of SeNPs synthesized in the presence of CBSE was evaluated since this is an important factor for the practical application of these nanoparticles in the food industry, particularly in providing nutritional supplements, as well as other possible applications in the medical field [30,35]. The stability of SeNPs obtained under optimal conditions was evaluated by analysing their visual appearance, UV-Vis spectra and antioxidant activity during 55 days under refrigerated conditions at 4 °C. Figure 6A shows the visual aspect of SeNPs after 0, 14, 35 and 55 days. A reddish-yellow colour was observed in these solutions due to the combination of the different nanoparticles obtained with different particle sizes (trigonal and amorphous synthesized Se-NPs), as already reported [66]. No apparent changes in SeNPs dispersions under refrigerated conditions were observed for 55 days, as SeNPs still were homogeneous without showing any precipitation, thus highlighting the excellent stability of these SeNPs synthesized with CBSE.

The UV-visible spectra obtained for SeNPs at different times are shown in Figure 6B. Following 55 days, no apparent changes in the absorption curve and optical properties were observed during storage, and no shift of the maximum absorption peak (276 nm) was noticed. However, from day 28 onward, a progressive decrease in peak intensity was observed, probably due to the degradation of some component of CBSE that can affect the intensity of the signal—although not producing any apparent modification in the SeNPs suspension — as already reported by other authors working with biologically-synthesized SeNPs [15,71].

The antioxidant capacity of CBSE and synthesized SeNPs with time also was evaluated by using the 2,2’-azino-bis(3-ethylbenzothiazoline-6-sulphonic acid) (ABTS) and ferric reducing antioxidant power (FRAP) spectrophotometric methods. According to previous reports, SeNPs exhibit excellent free radical scavenging capacities and antioxidant activities [72,73]. FRAP and ABTS values of 12.4 ± 0.2 and 28.6 ± 0.1 mg TE/g, respectively, were obtained for CBSE, highlighting the antioxidant activity of this natural extract. A significant increase (*p* = 0.05) in antioxidant performance was observed considering SeNPs giving rise to nanoparticles with high antioxidant potential (Figure 6C), indicating that selenium nanoparticles prepared using CBSE possess more antioxidant activity than the extract itself. A similar behaviour was reported for SeNPs synthesized with other natural extracts [51,53,74,75]. Moreover, as it can be seen in Figure 6C, the antioxidant capacity of SeNPs remained stable for 55 days and no significant differences (*p* > 0.05) were observed between samples analysed at different times, despite the decrease in UV-Vis intensity already discussed. These results are comparable to those reported for SeNPs synthesized using ascorbic acid in the presence of polysaccharides from *Catathelasma ventricosum* [54] where a high stability for SeNPs stored under refrigerated conditions for 60 days was observed. However, SeNPs synthesized in the presence of Gum–Arabic [69] and polysaccharide-protein complexes from *Corbicula flumine* [51] were stable just for 15 and 7 days, respectively, depending on the concentration of natural polysaccharides used. Therefore, microwave-assisted synthesized SeNPs using CBSE can be considered as highly stable antioxidant nanoparticles with time, opening possibilities for their potential application in several sectors.

## 3. Materials and Methods

### 3.1. Materials and Reagents

CBS waste was obtained from the production of chocolate and derivatives from a local company in Spain. CBS was washed twice with warm water at 30 °C and 1:1 ratio (*w*/*v*). Then, it was further dried at 30 °C for 24 h using a forced air oven, and ground to a particle size of 1 mm [76]. All chemicals were of analytical grade and they were purchased from Sigma–Aldrich (Madrid, Spain).

### 3.2. CBS Extract Preparation

Microwave-assisted extraction was used to obtain CBS extract (CBSE) by following a method previously optimized. Six grams of CBS particles were mixed with deionized water (0.04 g/mL) in a round-bottom flask and pH was adjusted to 2 by adding hydrochloric acid (1 M). Then, samples were heated in the microwave oven (Flexiwave, Milestone srl, Sorisole, Italy) at 100 °C for 5 min at 500 W of microwave power. A heating rate and stirring rate of 20 °C/min and 400 rpm were used, respectively. Subsequent to extraction, the extract was centrifuged at 4 °C and 5300 rpm for 20 min. Finally, the solid residue was separated and CBSE was stored at −20 °C until further use.

### 3.3. Quantitative Analysis of CBSE

Characterization of CBSE was performed, in triplicate, to determine its main constituents. Uronic acid content was determined by using the colorimetric sulfamate/3-phenylphenol method [77] at 520 nm using D-galacturonic acid as standard. Total polysaccharides content was measured by using the phenol-sulfuric acid method [78] at 490 nm using glucose as standard. The hydrolysable polyphenols content was evaluated using the Folin–Ciocalteu method [79] at 765 nm and TPC was expressed as mg of gallic acid equivalents per g of dried sample (mg GAE/g DW). Proteins content was determined according to the Bradford method using bovine serum albumin (BSA) as standard [80].

### 3.4. Synthesis of SeNPs

SeNPs were synthesized using Na_2_SeO_3_ as a selenium source. Different defined amounts of sodium selenite were diluted in 10 mL of deionized water and mixed with 50 mL of CBSE. The reaction solution was exposed to microwave irradiation using a Flexiwave microwave oven under constant stirring (400 rpm) and controlled time and microwave power conditions. Following reaction, the mixture was centrifuged at 5300 rpm and 4 °C for 15 min. Finally, the SeNPs suspensions were lyophilized and stored in the darkness until further characterization.

A rotatable and orthogonal central composite design (CCD) with three factors was used to evaluate the effect of the synthesis parameters, namely microwave power (W), irradiation time (min) and amount of Na_2_SeO_3_ (g), on crystalline size (nm) and *Z*-potential (eV) determined by using X-ray diffraction (XRD) and dynamic light scattering (DLS), respectively (Table 1). This experimental design contains a two-level full factorial design (2^3^ experiments), a star design (2 × 3 experiments) and a central point [81]. Each factor was varied at five levels (−α, −1, 0, +α, +1), with α = 1.68 and 9 replications of the central point also were performed.

Response surface methodology (RSM) was used for modelling and optimisation of SeNPs synthesis conditions to maximize the value of the *Z*-potential and minimize crystalline size. Experimental data were fitted by using the following second-order polynomial equation:*Y* = β*_o_* + Σβ*_i_*X_i_ + Σβ*_ii_*X_i_^2^ + Σβ*_ij_*X_i_X_j_(4)
where *Y* is the predicted response; β*_o_*, β*_i_*, β*_ii_*, and β*_ij_* are the regression coefficients for the intercept, linear, quadratic and interactive effect, respectively; and X*_i_* and X*_j_* are the actual values of the independent variables.

The suitability of the fitted models was evaluated by calculating coefficients of determination (R^2^), adjusted R^2^ values and lack of fit (*p*-value). ANOVA also was performed at the 95% confidence interval (α = 0.05). Three-dimensional surface plots were generated to evaluate interactions between synthesis factors and response variables.

### 3.5. Characterization of SeNPs

#### 3.5.1. X-ray Diffraction (XRD)

XRD patterns were recorded on a Bruker D8-Advance (Billerica, MA, USA) diffractometer in the 10–80° (2θ) range using filtered Cu Kα radiation (λ = 1.5406 Å). The average crystalline size of synthesized SeNPs was calculated by using the Debye–Scherrer equation:(5)$$D=\frac{K\lambda }{\beta \;\cos\theta }$$
where D is the crystal size, K is a constant whose value is approximately 0.9, λ is the wavelength of the X-ray, β is the full width at half maximum of the peak in radians, and θ is the Bragg’s diffraction angle in radians.

#### 3.5.2. Dynamic Light Scattering (DLS)

*Z*-potential values of the synthesized SeNPs were measured using a DLS Zetasizer Nano-ZS (Malvern Instruments, Worcestershire, UK). Dispersions of SeNPs in deionized water (1 mg/mL) were prepared and sonicated for 30 min before testing to obtain a good aqueous dispersion. Measurements were performed, in triplicate, using a disposable folded capillary cell.

#### 3.5.3. Optical Properties

UV-Vis spectrophotometry was used to confirm the formation of SeNPs, based on their surface plasmon resonance (SPR) signal, using a BioMate™ 3S spectrophotometer (Thermo Scientific, Waltham, MA, USA) in the 190–900 nm range.

#### 3.5.4. FTIR Analysis

FTIR spectra of samples were recorded, in triplicate, using an infrared spectrophotometer Bruker Analitik IFS 66/S (Ettlingen, Germany) in the 4000–500 cm^−1^ range (resolution 4 cm^−1^, 64 scans). This instrument was equipped with a KBr beam splitter and a DTGS detector, and OPUS software (Version 3.1) was used for analysis of spectra. Tests were performed in attenuated total reflectance (ATR) mode using a Golden Gate accessory with diamond crystal.

#### 3.5.5. Raman Spectroscopy

Raman spectra of SeNPs obtained under optimal conditions were recorded, in triplicate, using a Bruker RFS/100 FT-Raman spectrometer (Ettlingen, Germany) coupled with an optical microscope. An excitation laser line of 1064 nm (Nd-YAG) was used, to minimize problems of fluorescent samples.

#### 3.5.6. Transmission Electron Microscopy (TEM)

The morphology and size of SeNPs were evaluated with a JEOL JEM-1400 Plus TEM (Peabody, MA, USA) equipped with a Orius SC600 camera (Gatan, Pleasanton, CA, USA) for image acquisition at an accelerating voltage of 120 kV. SeNPs were prepared for TEM observation by dispersing them onto copper grids covered with a holey carbon film. The average particle size of SeNPs was obtained from TEM measurements of three replications.

#### 3.5.7. Stability and Antioxidant Activity of SeNPs

The stability of SeNPs during storage at 4 °C for 55 days also was studied. One milligram per millilitre suspensions of SeNPs obtained under optimal conditions were prepared using deionized water and sonicated for 30 min to obtain a good aqueous dispersion. Visual appearance of samples, changes in their UV-Vis spectra and antioxidant capacity were evaluated to estimate SeNPs stability with time. All tests were performed in triplicate.

Two different methods were used to evaluate the antioxidant performance of SeNPs: an ABTS (2, 2′ azinobis (3-ethylbenzthiazoline)-6-sulfonic acid) free radical decolourization assay, as an easy and accurate method, and a Ferric Reducing Antioxidant Power (FRAP) assay, which is very representative and useful in samples prepared in aqueous solutions [82]. These methods are based on the stabilization of free radicals and the ability of reducing an iron-complex, respectively. Each measurement was repeated 3 times.

The FRAP assay was determined as described by Martinez et al. [76]. The FRAP reagent was freshly prepared by mixing 300 mM acetate buffer (pH = 3.6), 10 mM 2,4,6-Tris(2-pyridyl)-s-triazine (TPTZ) and 20 mM ferric chloride in a 10:1:1 (*v*/*v*/*v*) mixing ratio. Then, 2 mL of the reagent and 200 µL of the testing sample solution were added to test tubes and incubated at 30 °C for 30 min. The absorbance was determined at 593 nm. (±)-6-Hydroxy-2,5,7,8-tetramethylchromane-2-carboxylic acid (Trolox) was used as standard and results were reported as mg of Trolox equivalents per gram of dried sample (mg TE/g DW) using a calibration curve obtained with Trolox in the 22–200 ppm range with 6 concentration points (R^2^ = 0.9987).

The ABTS assay also was performed as described by Martinez et al. [76]. The radical monocation of ABTS was generated by the reaction of the ABTS solution (7 mM) with 2.45 mM of potassium persulfate for 12 h at room temperature in the dark. The solution then was diluted with ethanol until obtaining an absorbance of 0.70 ± 0.02 at 734 nm. Aliquots of 150 μL of each sample were mixed with 2850 μL of the ABTS solution, and their absorbance was measured after 5 min. Trolox was used as standard and results also were reported as mg TE/g DW using a calibration curve in the 10–85 ppm range of Trolox with 6 concentration points (R^2^ = 0.9993).

### 3.6. Statistical Analysis

Statgraphics Centurion XVI (Statistical Graphics, Rockville, MD, USA) was used to generate and analyse the results of the CCD. The graphic analysis of the main effects and interactions between variables was used and the analysis of variance (ANOVA) was carried out. Differences between values were assessed based on confidence intervals by using the Tukey test at a p ≤ 0.05 significance level.

## 4. Conclusions

During this study, the efficiency of *Theobroma cacao* L. bean shell extract as a stabilizing and capping agent in the green microwave-assisted synthesis of Se nanoparticles has been proven, for the first time. RSM was successfully applied to model SeNPs synthesis, resulting in the production of spherical antioxidant nanoparticles with a particle size of 1–3 nm, clearly being lower in size than previously reported SeNPs synthesized by other authors, and directly influencing its biological properties. Both trigonal and amorphous synthesized SeNPs were found according to XRD and Raman results. The obtained SeNPs showed more antioxidant activity than CBSE alone, and they were well stabilized by CBSE biomolecules preventing aggregation for around two months under refrigerated conditions, showing great potential for many applications in different fields, such as the food and medical industries. Further studies on these applications should be performed to finally assess their suitability for these specific applications.

## Figures and Tables

**Figure 1 molecules-24-04048-f001:**
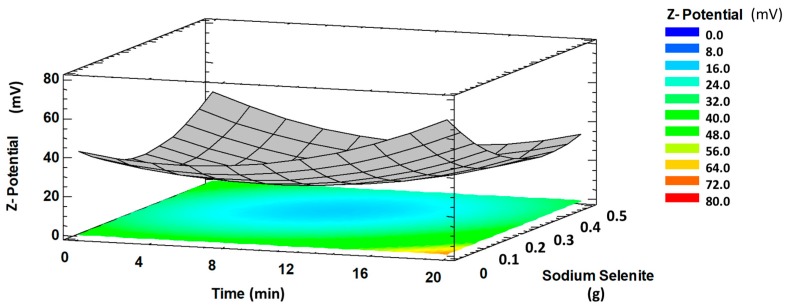
Response surface plot showing significant interaction time (min) versus Na_2_SeO_3_ (g) on *Z*-potential (absolute values).

**Figure 2 molecules-24-04048-f002:**
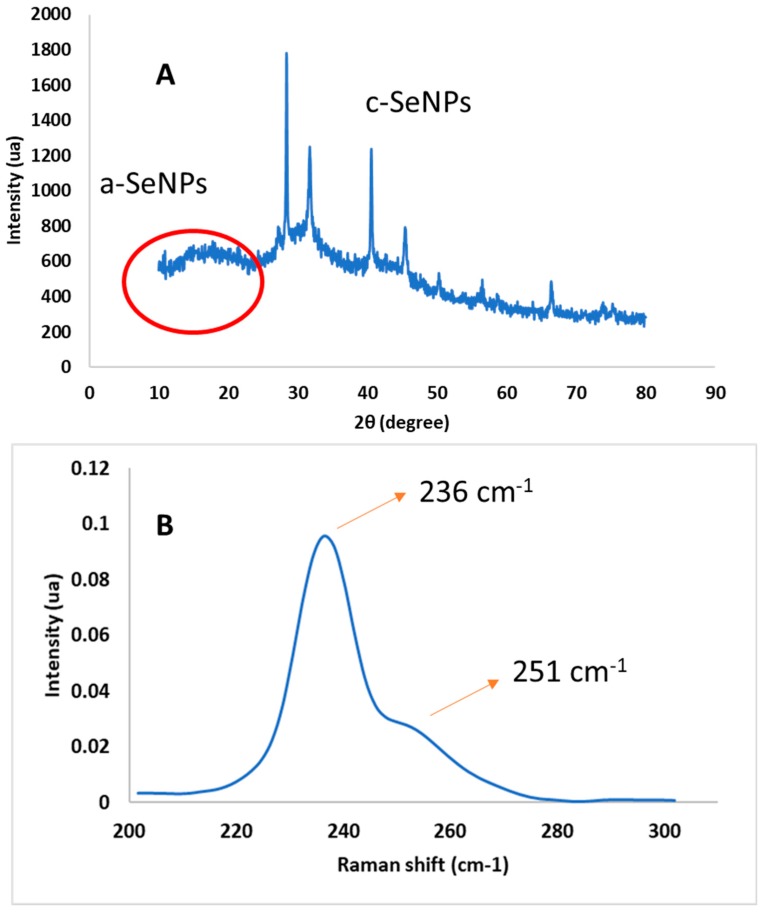
XRD pattern (**A**) and Raman spectrum (**B**) of SeNPs obtained under optimal conditions.

**Figure 3 molecules-24-04048-f003:**
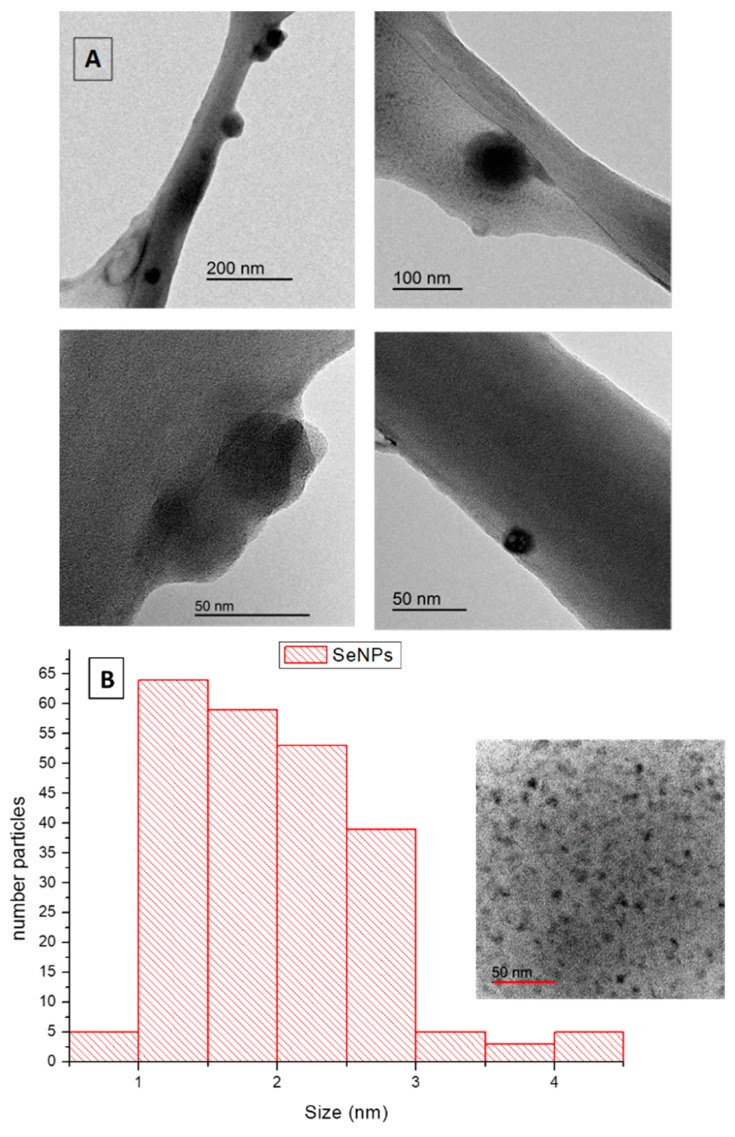
TEM images (**A**) and size distribution (**B**) of SeNPs obtained under optimal conditions.

**Figure 4 molecules-24-04048-f004:**
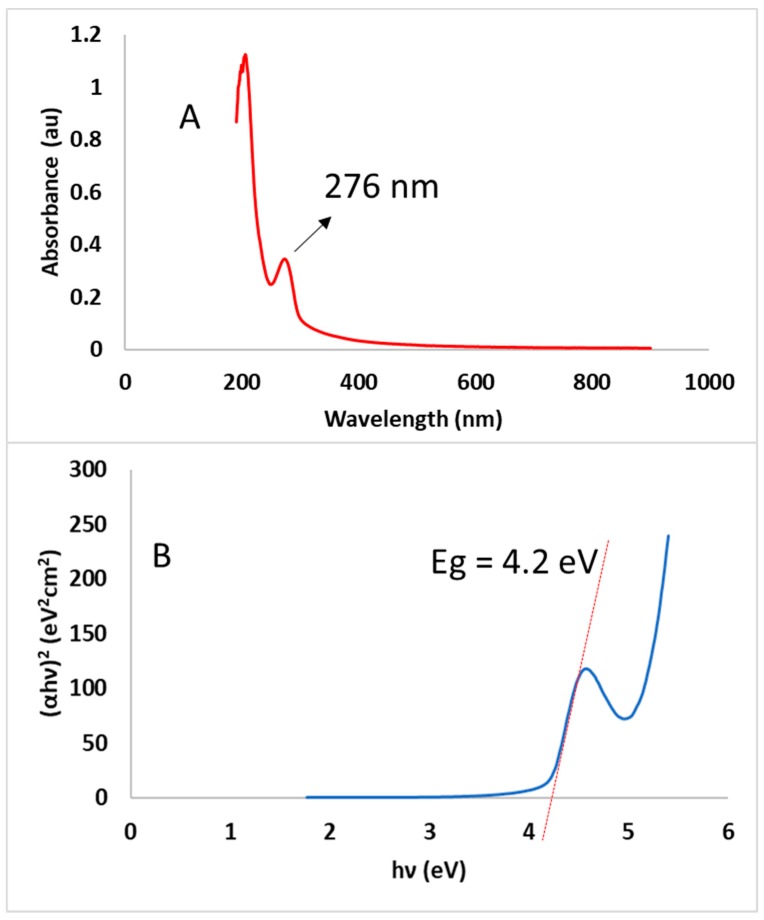
UV-Vis spectrum (**A**) and band gap energy (**B**) of SeNPs obtained under optimal conditions.

**Figure 5 molecules-24-04048-f005:**
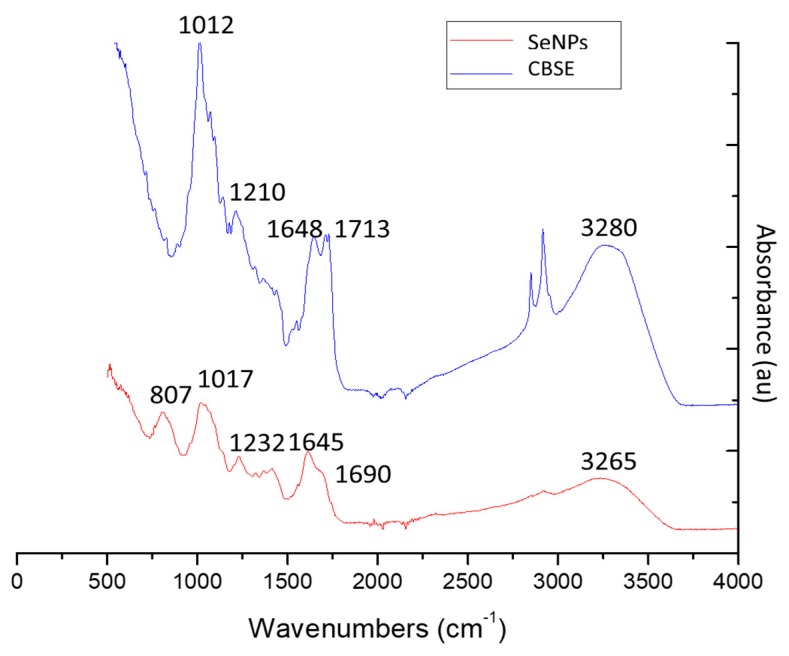
FTIR spectra of CBSE and SeNPs obtained under optimal conditions.

**Figure 6 molecules-24-04048-f006:**
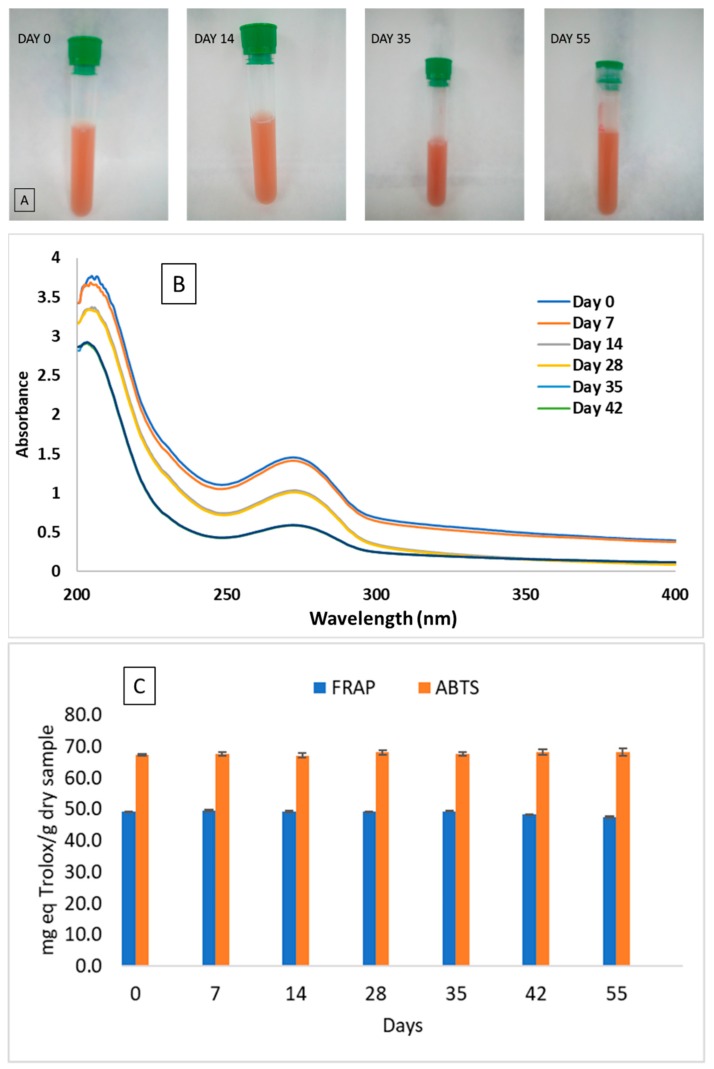
Visual appearance (**A**), UV-Vis spectra (**B**) and antioxidant activity (**C**) of SeNPs after 0, 14, 35 and 55 days stored at 4 °C.

**Table 1 molecules-24-04048-t001:** Chemical characterization of CBSE used for SeNPs synthesis (*n* = 3, mean ± SD).

Constituents	Content
Uronic acid (mg GlcA/g)	82.7 ± 6.3
Total phenolics (mg GAE/g)	23.2 ± 0.4
Total polysaccharides (mg Glu/g)	300.8 ± 2.5
Proteins (mg BSA/g)	170.3 ± 3.1

**Table 2 molecules-24-04048-t002:** Central composite design matrix and response values obtained for the synthesis of SeNPs.

Run	Time (min)	Power (W)	Na_2_SeO_3_ (g)	Average Crystalline Size (nm) *	*Z*-Potential (mV) **
1	10.0	600.0	0.275	53.1	−19.27
2	15.0	400.0	0.150	43.8	−33.60
3	1.6	600.0	0.275	46.9	−26.07
4	10.0	263.6	0.275	48.5	−24.67
5	10.0	600.0	0.065	48.9	−31.67
6	18.4	600.0	0.275	46.5	−25.63
7	15.0	800.0	0.150	40.9	−36.47
8	10.0	600.0	0.275	45.9	−14.57
9	5.0	800.0	0.150	43.4	−28.53
10	10.0	600.0	0.275	49.9	−18.50
11	10.0	600.0	0.275	47.5	−15.67
12	5.0	800.0	0.400	58.9	−25.03
13	15.0	400.0	0.400	66.6	−25.70
14	5.0	400.0	0.400	58.0	−27.30
15	10.0	600.0	0.275	48.6	−14.77
16	10.0	600.0	0.275	51.5	−14.73
17	10.0	936.4	0.275	45.6	−26.97
18	15.0	800.0	0.400	58.1	−26.77
19	5.0	400.0	0.150	43.8	−22.27
20	10.0	600.0	0.275	49.1	−16.30
21	10.0	600.0	0.485	66.2	−22.50
22	10.0	600.0	0.275	49.2	−17.90
23	10.0	600.0	0.275	48.9	−15.13

* Determined by X-ray diffraction (XRD) using the Debye–Scherrer equation; ** Determined by dynamic light scattering (DLS).

**Table 3 molecules-24-04048-t003:** ANOVA results for response surface quadratic models in SeNPs synthesis.

Source	Sum of Squares	DF	Mean Square	*F*-Value	*p*-Value
**Size**					
A	1.57	1	1.57	0.35	0.5688
B	18.16	1	18.16	4.09	0.0777
C	715.34	1	715.34	161.25	0.0001 ***
AA	7.17	1	7.17	1.62	0.2393
AB	17.73	1	17.73	4.00	0.0806
AC	13.03	1	13.03	2.94	0.1249
BB	4.77	1	4.77	1.08	0.3300
BC	2.48	1	2.48	0.56	0.4765
CC	158.74	1	158.74	35.78	0.0003 ***
Lack of fit	55.06	5	11.01	2.48	0.1215
Pure error	35.49	8	4.44		
Cor. Total	1030.31	22			
R^2^	0.9121				
Adj R^2^	0.8512				
CV (%)	2.64				
***Z*-Potential**					
A	25.52	1	25.52	7.92	0.0227 *
B	10.19	1	10.19	3.16	0.1132
C	72.62	1	72.62	22.53	0.0015 **
AA	211.89	1	211.89	65.75	0.0001 ***
AB	0.00	1	0.00	0.00	0.9924
AC	45.74	1	45.74	14.19	0.0055 **
BB	210.66	1	210.66	65.36	0.0001 ***
BC	13.34	1	13.34	4.14	0.0763
CC	265.59	1	265.59	82.41	0.0001 ***
Lack of fit	35.21	5	7.04	2.19	0.1558
Pure error	25.78	8	3.22		
Cor. Total	907.21	22			
R^2^	0.9327				
Adj R^2^	0.8862				
CV (%)	2.16				

A: extraction time; B: microwave power; C: sodium selenite amount; * Significant, *p* < 0.05, ** Very significant, *p* < 0.01, *** Highly significant, *p* < 0.001.

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
