# Peer review of "Microwave-Assisted Green Synthesis and Antioxidant Activity of Selenium Nanoparticles Using Theobroma cacao L. Bean Shell Extract"

_molecules, 2019, doi:10.3390/molecules24224048_

Round 1

Reviewer 1 Report

Mellinas and co-workers are presented a controlled synthesis of SeNPs. Although the idea of using similar natural sources for NP synthesis is not new, the statistical approach is relevant. I recommend the publication of the article. My minor comments are below,

Equations 1&2 given here have 12 decimal place significance. I do not think it is possible here. Consider the writing the equations with correct significant figures. Authors use Z-potential and potential-Z at times. Are they any different?

Figure 1. The Z-potentials are negative values in the text but why this axis is reading positive?

Figure 6. The absorption spectrum showing a peak at UV range but the photographs showing a pale yellow color (visible region). Is there any peaks around 400-550 nm or it's from scattering? or some other molecule absorbing?

Author Response

Comments and Suggestions for Authors

Mellinas and co-workers are presented a controlled synthesis of SeNPs. Although the idea of using similar natural sources for NP synthesis is not new, the statistical approach is relevant. I recommend the publication of the article. My minor comments are below,

Answer: We thank the reviewer for his/her positive comments.

Equations 1&2 given here have 12 decimal place significance. I do not think it is possible here. Consider the writing the equations with correct significant figures. Authors use Z-potential and potential-Z at times. Are they any different?

Answer: The significant figures of equations 1 and 2 have been corrected. We have also corrected and unified the term Z-potential all along the text and in Figure 1.

Figure 1. The Z-potentials are negative values in the text but why this axis is reading positive?

Answer: As explained in the text (section 2.3.4) “the sign of Z-potential gives information about the compounds that stabilize SeNPs but the absolute value refers to the degree of stabilization under the studied conditions due to Van der Walls interactions When we applied the RSM to the results our objective was to obtain Se-NPs with high stability and low size. For this reason, we used the absolute value of Z-potential (knowing that the overall values obtained here were all negative) to analyse the experimental design. Figure 1 shows the significant interactions between time and concentration of the metal precursor, and for plotting purposes we decided to use absolute values of Z-potential. Figure 1 caption has been modified to clarify this point.

Figure 6. The absorption spectrum showing a peak at UV range but the photographs showing a pale yellow color (visible region). Is there any peaks around 400-550 nm or it's from scattering? or some other molecule absorbing?

Answer: The peak with maximum absorbance of Se-NPs synthesized in this work was observed at the UV-region (276 nm), showing the optical absorption spectrum a large blue shift of the band gap energy for the small sized Se-NPs compared to its bulk counterpart (775 nm) due to the quantum confinement effect. At larger particles size, a red shift in optical spectra is expected. As a result, the combination of all the different nanoparticles obtained in this work with different particle sizes (trigonal and amorphous synthesized Se-NPs) resulted in an apparent reddish yellow colour observed in Figure 6.

Reviewer 2 Report

The manuscript “Microwave-Assisted Green Synthesis and Antioxidant Activity of Selenium Nanoparticles Using Theobroma Cacao L. Bean Shell Extract submitted to Molecules demonstrates the green synthesis of selenium nanoparticles using Theobroma Cacao L. Bean Shell Extract. The objectives of the research are well-founded, as currently nanotechnology is a fast-developing industry, posing substantial impacts on economy, society and environment. The vast production, use, and disposal of nanomaterials may have long lasting effect on the environment. In this context, the green synthesis of nanomaterials is highly desirable. In this study, smaller size crystalline Se nanoparticles are biosynthesized in a microwave-assisted method using the shell extract. Indeed the application of RSM method in the study to determine the optimal conditions for the synthesis of NPs will be very helpful for future studies. The manuscript is well designed and the characterizations of as-prepared samples are sufficient which is performed using various techniques. Therefore, the manuscript is suitable for acceptance after the authors address the following concerns.

Replace the images of TEM with more clear scale bar.

Correct the sentences in the following lines 52-54, 58, 60, 84, 227, and also check the manuscript thoroughly for other grammatical mistakes.

The authors should include few references about the importance of green synthesis of nanomaterials such as,

Dalton Transactions 44.21 (2015): 9709-9717,

Dalton Transactions 47.35 (2018): 11988-12010

Author Response

Comments and Suggestions for Authors

The manuscript “Microwave-Assisted Green Synthesis and Antioxidant Activity of Selenium Nanoparticles Using Theobroma Cacao L. Bean Shell Extract submitted to Molecules demonstrates the green synthesis of selenium nanoparticles using Theobroma Cacao L. Bean Shell Extract. The objectives of the research are well-founded, as currently nanotechnology is a fast-developing industry, posing substantial impacts on economy, society and environment. The vast production, use, and disposal of nanomaterials may have long lasting effect on the environment. In this context, the green synthesis of nanomaterials is highly desirable. In this study, smaller size crystalline Se nanoparticles are biosynthesized in a microwave-assisted method using the shell extract. Indeed the application of RSM method in the study to determine the optimal conditions for the synthesis of NPs will be very helpful for future studies. The manuscript is well designed and the characterizations of as-prepared samples are sufficient which is performed using various techniques. Therefore, the manuscript is suitable for acceptance after the authors address the following concerns.

Answer: We thank the reviewer for his/her positive comments.

Replace the images of TEM with more clear scale bar.

Answer: We have highlighted the scale bar in Figure 3b according to the reviewer comment to have a more clearer vision.

Correct the sentences in the following lines 52-54, 58, 60, 84, 227, and also check the manuscript thoroughly for other grammatical mistakes.

Answer: These sentences have been corrected (in red in the revised version) and the whole manuscript has been carefully reviewed in grammar and spelling.

The authors should include few references about the importance of green synthesis of nanomaterials such as,

Dalton Transactions 44.21 (2015): 9709-9717,

Dalton Transactions 47.35 (2018): 11988-12010

Answer: Both references have been included in the Introduction section.